# $K^*$ Search Over Orbit Space for Top-k Planning

## Michael Katz, Junkyu Lee

IBM T.J. Watson Research Center
1101 Kitchawan Rd, Yorktown Heights, NY 10598, USA
{michael.katz1, junkyu.lee}@ibm.com

## Abstract

Top-k planning, the task of finding $k$ top-cost plans, is a key formalism for many planning applications and $K^*$ search is a well-established approach to top-k planning. The algorithm iteratively runs $A^*$ search and Eppstein's algorithm until a sufficient number of plans is found. The performance of $K^*$ algorithm is therefore inherently limited by the performance of $A^*$, and in order to improve $K^*$ performance, that of $A^*$ must be improved. In cost-optimal planning, orbit space search improves $A^*$ performance by exploiting symmetry pruning, essentially performing $A^*$ in the orbit space instead of state space. In this work, we take a similar approach to top-k planning. We show theoretical equivalence between the goal paths in the state space and in the orbit space, allowing to perform $K^*$ search in the orbit space instead, reconstructing plans from the found paths in the orbit space. We prove that our algorithm is sound and complete for top-k planning and empirically show it to achieve state-of-the-art performance, overtaking all existing to date top-k planners. The code is available at https://github.com/IBM/kstar.

## Introduction

Top-k planning is essential to many planning applications, including malware detection (Boddy et al. 2005), hypothesis generation (Sohrabi et al. 2016), scenario planning (Sohrabi et al. 2018), and machine learning pipeline generation (Katz et al. 2020). There are currently three main approaches to top-k planning. The last one chronologically is based on a so-called symbolic search (Speck, Mattmüller, and Nebel 2020). Another approach to top-k planning is called Forbid Iterative (FI). It is based on iteratively reformulating the input task, excluding the previously found plans from the task plans space (Katz et al. 2018b). The third approach is based on a $K^*$ search (Aljazzar and Leue 2011). In planning, it was first implemented within a planner for an SPPL language (Riabov and Liu 2006), with some simplifications (Riabov, Sohrabi, and Udrea 2014). Later, a similar variant was implemented within a PDDL planner (Katz et al. 2018b), and most recently the original variant of Aljazzar and Leue (2011) was implemented on top of a PDDL planner (Lee, Katz, and Sohrabi 2023). That last variant significantly improves the performance of $K^*$ search, compared to

the previous one and for the first time allows for using inconsistent heuristics. It works in two interchanging phases, $A^*$ search and Eppstein's k shortest paths algorithm (Eppstein 1998) on the developed search space. Thus, the performance of $K^*$ algorithm is inherently limited by that of $A^*$ and in order to improve $K^*$ performance, one must improve $A^*$.

In cost-optimal planning, the issue of $A^*$ performance was tackled, among other, by using search space pruning techniques. Two main such techniques are partial order reduction (Wehrle and Helmert 2012) and symmetry based search pruning (Pochter, Zohar, and Rosenschein 2011; Domshlak, Katz, and Shleyfman 2012). For the latter, two search algorithms were suggested, DKS (Domshlak, Katz, and Shleyfman 2012) and orbit space search (OSS) (Domshlak, Katz, and Shleyfman 2015). While DKS performs the search in problem state space, pruning states that deem symmetric to the previously encountered ones, OSS is essentially performing $A^*$ search in a so-called orbit space instead of the state space. In the orbit space, nodes correspond to equivalence classes of states, with the equivalence relation based on detecting states as symmetric. In both algorithms, the symmetry detection is done via so-called canonical states. While there is no clear dominance of one of the methods over the other, DKS requires storing two states per search node, the actual state and the canonical one. OSS, on the other hand, requires only storing the canonical state. Thus, in principle, it should have a smaller memory consumption.

In this work, we take a similar approach to top-k planning. First, as $K^*$ requires transforming the input planning task into a task with a single goal state, we propose a transformation into a planning task with a single goal state that preserve the symmetries of the input task. Next, focusing on the orbit space search, we establish the theoretical grounds for performing $K^*$ search in the orbit space. We do that by showing the equivalence between the goal paths in the state space and the orbit space. These goal paths in the orbit space can then be cast into actual plans. We then prove that our suggested algorithm is sound and complete for top-k planning. Last, we perform an extensive empirical evaluation of the proposed algorithm with a variety of admissible heuristics, both consistent and inconsistent, and comparing to all existing top-k planners. Our evaluation shows the new approach to be competitive with the current state-of-the-art.

# Background

This section introduces necessary concepts in top-$k$ planning, $K^*$ search algorithm for top-$k$ planning, and OSS for cost optimal planning.

## Top-$k$ Classical Planning

A *planning task* $\Pi = \langle \mathcal{V}, \mathcal{O}, s_0, s_\star \rangle$ in SAS$^+$ formalism (Bäckström and Nebel 1995) consists of a finite set of finite-domain *state variables* $\mathcal{V}$, a finite set of *actions* $\mathcal{O}$, an *initial state* $s_0$, and the *goal* $s_\star$. Each variable $v \in \mathcal{V}$ is associated with a finite domain $dom(v)$ of values. An assignment of a value $d \in dom(v)$ to a variable $v \in \mathcal{V}$ denoted by a pair $\langle v, d \rangle$ is called *fact*, and the set of all facts is denoted by $F$. A *partial assignment* $p$ maps a subset of variables $vars(p) \subseteq \mathcal{V}$ to values in their domains. For a variable $v \in \mathcal{V}$ and partial assignment $p$, the value of $v$ in $p$ is denoted by $p[v]$ if $v \in vars(p)$ and we say $p[v]$ is *undefined* otherwise. A full assignment $s$ is called a *state*, and the set of all states is denoted by $\mathcal{S}$. State $s$ is *consistent* with a partial assignment $p$ if they agree on all variables in $vars(p)$, denoted by $p \subseteq s$. Each action $o$ in $\mathcal{O}$ is a pair $\langle pre(o), eff(o) \rangle$, where $pre(o)$ and $eff(o)$ are partial assignments called *precondition* and *effect*. Further, $o$ has an associated non-negative cost denoted by $cost(o) \in \mathbb{R}^{0+}$. An action $o$ is applicable in state $s$ if $pre(o) \subseteq s$. Applying $o$ in $s$ results in a state denoted by $s[\![o]\!]$, where $s[\![o]\!][v] = eff(o)[v]$ for all $v \in vars(eff)$ and $s[\![o]\!][v] = s[v]$ for all other variables. An action sequence $\pi = \langle o_1, \ldots, o_n \rangle$ is applicable in state $s$ if there are states $\langle s_0, \ldots, s_n \rangle$ such that $o_i$ is applicable in $s_{i-1}$ and $s_{i-1}[\![o_i]\!] = s_i$ for $0 \le i \le n$. We denote $s_n$ by $s[\![\pi]\!]$. An action sequence with $s_\star \subseteq s_0[\![\pi]\!]$ is called a *plan*. The cost of a plan $\pi$, denoted by $cost(\pi)$ is the summed cost of the actions in the plan. The set of all plans is denoted by $\mathcal{P}_\Pi$, and an *optimal* plan is a plan in $\mathcal{P}_\Pi$ with the lowest cost.

**Example 1 (Gripper Task)** *We will use a famous gripper task introduced by Jana Koehler as a running example throughout the paper. There is a robot* R *with two grippers* $l$ *and* $r$, *and each can carry a ball. The goal is to move four balls,* $b_1, b_2, b_3, b_4,$ *from room* A *to room* B.

- *A state can be represented by seven* SAS$^+$*variables: one variable* R *with* $dom(R) = \{A, B\}$ *for encoding the location of the robot, four variables,* $\{b_i | i \in [1..4]\}$ *with* $dom(b_i) = \{A, B, R\}$ *for encoding the location of the four balls, and two variables,* $l$ *and* $r$ *for grippers with domain* $\{E, b_1, b_2, b_3, b_4\}$ *for encoding the object a gripper holds, where* E *represents holding nothing.*
- *For brevity, we denote a state with five letters, comprising the location of the robot and four balls. Namely,* ARLBB *stands for a state with facts:* $\langle R, A \rangle$, $\langle b_1, r \rangle$, $\langle b_2, l \rangle$, $\langle b_3, B \rangle$, $\langle b_4, B \rangle$, $\langle l, b_2 \rangle$, *and* $\langle r, b_1 \rangle$.
- *Actions are* pick, drop, *and* move *for manipulating the balls and moving between rooms, abbreviated as follows.* P1LA *denotes a* pick *action taking* $b_1$ *with the left gripper in room A,* D2RB *denotes a* drop *action dropping* $b_2$ *from the right gripper in room B,* MAB *denotes a* move *action moving from room A to B, etc.*

Next, we present the *top-$k$ planning problem*, as defined by Sohrabi et al.; Katz et al. (2016; 2018b).

**Definition 1 (top-$k$ planning problem)** *Given a classical planning task* $\Pi$ *and a natural number* $k,$ *top-$k$ planning problem is finding a set of plans* $P \subseteq \mathcal{P}_\Pi$ *satisfying the following properties.*

1. *For all plans* $\pi \in P$, *if there exists a plan* $\pi' \in \mathcal{P}_\Pi$ *such that* $C(\pi') < C(\pi)$, *then* $\pi' \in P$,
2. $|P| \le k$, *and if* $|P| < k$, *then* $P = \mathcal{P}_\Pi$.

We say a top-$k$ planning problem $\langle \Pi, k \rangle$ is solvable if $|P| = k$ and unsolvable if $|P| < k$. Note that cost-optimal planning is a special case of top-$k$ planning for $k = 1$.

## $K^*$ Search for Top-$k$ Planning

Given a top-$k$ planning problem $\langle \Pi, k \rangle$, $K^*$ applies Eppstein's algorithm ($EA$) to the search graph revealed by $A^*$ until the task is solved or proven to be unsolvable. In this section, we only review necessary concepts and refer to Aljazzar and Leue (2011) and Eppstein (1998) for details.

**Single Goal State Reformulation of Planning Tasks** The fact that any state consistent to $s_\star$ is a goal state in a planning task $\Pi = \langle \mathcal{V}, \mathcal{O}, s_0, s_\star \rangle$ calls for a single goal state reformulation since $K^*$ assumes a single terminal node in the search graph. Katz et al. (2018b) showed such a reformulated task $\Pi^g = \langle \mathcal{V}^g, \mathcal{O}^g, s_0^g, s_\star^g \rangle$, where $\mathcal{V}^g = \mathcal{V} \cup \{v_g\}$ with a binary indicator variable $v_g$ for reaching a goal state, $\mathcal{O}^g = \{\langle pre(o) \cup \langle v_g, 0 \rangle, eff(o) \rangle | o \in \mathcal{O}\} \cup \{o_g\}$ with a zero cost goal-achieving action $o_g$ such that $pre(o_g) = s_\star \cup \{\langle v_g, 0 \rangle\}$ and $eff(o_g) = \{\langle v_i, t[v_i] \rangle | v_i \in vars(t)\} \cup \{\langle v_g, 1 \rangle\}$ for an arbitrary full state $t$, $s_0^g = s_0 \cup \{\langle v_g, 0 \rangle\}$, and $s_\star^g = eff(o_g)$.

**Implicit Path Representation of Eppstein's Algorithm** $A^*$ explores a state transition graph $\mathcal{T}_\Pi = \langle \mathcal{S}, \mathcal{E} \rangle$ comprised of nodes associated with states $\mathcal{S}$ and edges $\mathcal{E}$, induced by operators $\mathcal{O}$, namely, $\{\langle s, s[\![o]\!]; o \rangle | pre(o) \in s \ \forall s \in \mathcal{S}, \forall o \in \mathcal{O}\}$. Denoting by $G_{A^*}$ an explicit search graph revealed by $A^*$, $EA$ uses an implicit path representation relative to the shortest path tree $T_{A^*}$ using a sequence of "side-tracked" edges (STE), the edges of $G_{A^*}$ that are not in $T_{A^*}$. For each STE $(u, v)$, we can compute the deviation cost $\delta(u, v)$ against the cost of the incoming edge toward $v$ in $T_{A^*}$ by $\delta(u, v) = g(u) + cost(o) - g(v)$ iff $v = u[\![o]\!]$. We denote an arbitrary goal reaching path from $s_0^g$ in $G_{A^*}$ by $\rho_{A^*}(s_0^g, s_\star^g)$, and the unique path from $u$ to $v$ in $T_{A^*}$ by $\rho_{A^*}^*(u, v)$ if it exists. Then, any $\rho_{A^*}(s_0^g, s_\star^g)$ can be uniquely represented by an ordered sequence of STEs, denoted by SIDETRACKS$(\rho_{A^*}(s_0^g, s_\star^g)) = \langle (u_1, v_1), \ldots, (u_q, v_q) \rangle$, where an STE closer to $s_\star^g$ appears earlier in the sequence. Namely, $\rho_{A^*}(s_0^g, s_\star^g)$ can be reconstructed from SIDETRACKS$(\rho_{A^*}(s_0^g, s_\star^g))$ by

$$\rho_{A^*}(s_0^g, s_\star^g) = \rho_{A^*}^*(s_0^g, u_q) \circ [\circ_{i=q}^2 \{(u_i, v_i) \circ \rho_{A^*}^*(v_i, u_{i-1})\}] \\ \circ (u_1, v_1) \circ \rho_{A^*}^*(v_1, s_\star^g),$$

where $\circ$ concatenates edges and paths from left to right.

$EA$ performs Dijkstra's algorithm over its own search graph, called *path graph* $P(G_{A^*})$ in which each node represents SIDETRACKS$(\rho_{A^*}(s_0^g, s_\star^g))$ with the total deviation

cost $\sum_{i=1}^{q} \delta(u_i, v_i)$. Therefore, $k$-shortest paths are found in the order of their costs.

## Orbit Space Search for Cost Optimal Planning

Symmetry breaking techniques (Pochter, Zohar, and Rosenschein 2011; Domshlak, Katz, and Shleyfman 2012; Shleyfman et al. 2015) significantly improved heuristic forward state space search for cost optimal classical planning. Given a planning task $\Pi$, and its state transition graph $\mathcal{T}_\Pi$, a graph automorphism in the automorphism group of $\mathcal{T}_\Pi$, denoted by $Aut(\mathcal{T}_\Pi)$, permutes states while preserving the adjacency between states. Any subgroup of $Aut(\mathcal{T}_\Pi)$ partitions the state space by orbits, which offers opportunity for pruning the search space. In this section, we introduce necessary concepts in OSS.

## Structural Symmetries and Canonical States

Since state transition graphs are implicit, we find graph automorphisms from the syntactic structure of planning tasks, called structural symmetries (Shleyfman et al. 2015), and follow the definition shown in Sievers et al. (2017).

**Definition 2 (Structural Symmetry)** *Given a planning task $\Pi = \langle \mathcal{V}, \mathcal{O}, s_0, s_\star \rangle$, a structural symmetry is a permutation $\sigma : \mathcal{V} \cup F \cup \mathcal{O} \to \mathcal{V} \cup F \cup \mathcal{O}$ with the following properties.*

1. *$\sigma(\mathcal{V}) = \mathcal{V}$ and $\sigma(F) = F$ such that $\sigma(\langle v, d \rangle) = \langle v', d' \rangle$ implies $v' = \sigma(v)$,*
2. *$\sigma(\mathcal{O}) = \mathcal{O}$ such that for $o \in \mathcal{O}$, $\sigma(pre(o)) = pre(\sigma(o))$, $\sigma(eff(o)) = eff(\sigma(o))$, and $cost(\sigma(o)) = cost(o)$,*
3. *$\sigma(s_\star) = s_\star$,*

*where $\sigma(\{x_1, \ldots, x_n\}) := \{\sigma(x_1), \ldots, \sigma(x_n)\}$.*

Note that the third property ensures that a structural symmetry $\sigma$ *stabilizes* the goal. In practice, a set of structural symmetries can be detected by using tools for finding automorphisms of colored digraphs, such as Bliss (Junttila and Kaski 2007). These structural symmetries are the generators of the symmetry group $\Gamma$, a subgoup of $Aut(\mathcal{T}_\Pi)$.

Given two states $s$ and $t$, it is crucial to quickly check whether $s$ and $t$ are symmetric or not for speeding up search. Unfortunately, finding $\sigma \in \Gamma$ for $s = \sigma(t)$ is PSPACE-hard (Shleyfman and Jonsson 2021). Instead of finding such $\sigma$, Pochter, Zohar, and Rosenschein (2011) proposed to use *canonical form* (Emerson and Sistla 1996), which we call canonical states.

**Definition 3 (Canonical States)** *Given a planning task $\Pi = \langle \mathcal{V}, \mathcal{O}, s_0, s_\star \rangle$ over the states $\mathcal{S}$, and a group $\Gamma$ of structural symmetries for $\Pi$, canonical state $C_\Gamma(s)$ of $s \in \mathcal{S}$ is a state generated by a mapping $C_\Gamma : \mathcal{S} \to \mathcal{S}$ such that for all $s, t \in \mathcal{S}$, $C_\Gamma(s) = C_\Gamma(t)$ iff there exists $\sigma \in \Gamma$ such that $s = \sigma(t)$.*

The common choice of $C_\Gamma$ implements a greedy search that outputs local lexicographically minimum states since finding the global minimum is NP-hard (Luks 1993). Nevertheless, missing to match structurally symmetric states only skips pruning symmetric states and it doesn't forfeit the soundness and completeness of search (Domshlak, Katz, and Lefler 2012).

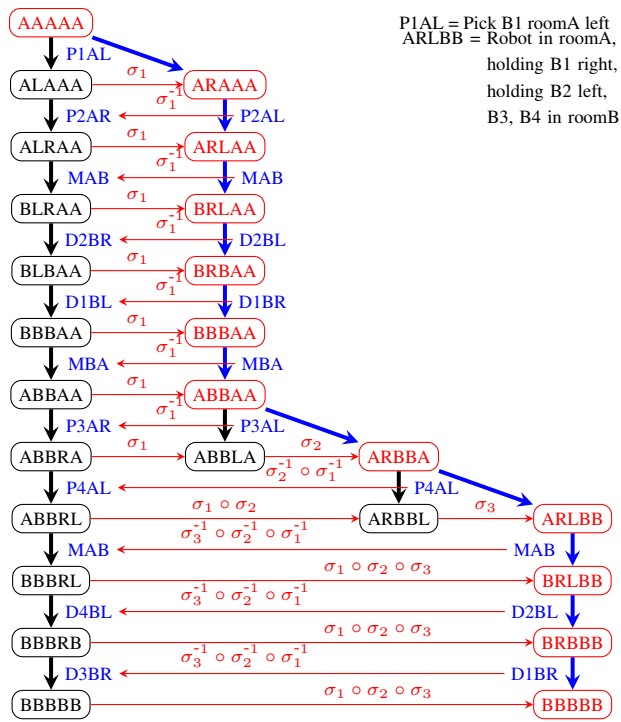

Figure 1: Gripper example: TRACE-FORWARD algorithm.

## Example 2 (Sturctural Symmetries in Gripper Task)

*Using the tools mentioned above, we can find a structural symmetries group with four generators as follows. Each of these structural symmetries is of order 2, meaning if $\sigma(x) = y$ then also $\sigma(y) = x$ must hold, so for brevity we describe only one of these. For unmentioned elements $x$, we have $\sigma(x) = x$.*

1. *Permuting the left gripper $l$ and the right gripper $r$: $\sigma_{lr}(l) = r$, $\forall d \in dom(l)$, $\sigma_{lr}(\langle l, d \rangle) = \langle r, d \rangle$, $\forall X \in \{1..4\}$, $\forall Y \in \{A, B\}$, $\sigma_{lr}(PXYL) = PXYR$, $\forall X \in \{1..4\}$, $\forall Y \in \{A, B\}$, $\sigma_{lr}(DXYL) = DXYR$,*
2. *permuting balls $b_1$ and $b_2$: $\sigma_{12}(b_1) = b_2$, $\forall d \in dom(b_1)$, $\sigma_{12}(\langle b_1, d \rangle) = \langle b_2, d \rangle$, $\forall v \in \{l, r\}$, $\sigma_{12}(\langle v, b_1 \rangle) = \langle v, b_2 \rangle$, $\forall X \in \{A, B\}$, $\forall Y \in \{L, R\}$, $\sigma_{12}(P1XY) = P2XY$, $\forall X \in \{A, B\}$, $\forall Y \in \{L, R\}$, $\sigma_{12}(D1XY) = D2XY$,*
3. *$\sigma_{23}$, permuting balls $b_2$ and $b_3$, similarly to the above,*
4. *$\sigma_{34}$, permuting balls $b_3$ and $b_4$, similarly to the above.*

**Orbit Space Search** Unlike $A^*$, which must expand all the symmetric states, OSS prunes them and explores a compact canonical state transition graph defined as follows.

**Definition 4 (Canonical State Transition Graph)** *Given a planning task $\Pi$ and its state transition graph $\mathcal{T}_\Pi(\mathcal{S}, \mathcal{E})$ over states $\mathcal{S}$ with labeled state transitions $\mathcal{E}$ induced by applicable operators in $\mathcal{O}$, we define a canonical state transition graph $\mathcal{T}'_\Pi(\mathcal{S}', \mathcal{E}')$ relative to a structural symmetry group $\Gamma$ such that:*

1. *$\mathcal{S}' = \{\sigma(s) | s \in \mathcal{S}, C_\Gamma(s) = \sigma(s)\}$,*

2. $\mathcal{E}' = \{\langle \sigma_i(s_i), \sigma_j\sigma_i(s_j); \sigma_i(o)\rangle | \langle s_i, s_j; o\rangle \in \mathcal{E}, C_\Gamma(s_i) = \sigma_i(s_i), C_\Gamma(s_j) = \sigma_j(s_j)\}$,

*where $\sigma_i \in \Gamma$ maps state $s_i$ to its canonical state $C_\Gamma(s_i)$, and the transition labels are induced by operators applicable in canonical states.*

OSS explores $\mathcal{T}'_\Pi(\mathcal{S}', \mathcal{E}')$ by $A^*$ and traverses an explicit search graph $G_\text{OSS}$, which replaces any generated state $s$ with its canonical state $C_\Gamma(s)$. Denoting by $\rho_\text{OSS}(u, v)$ a path from $u$ to $v$ in $G_\text{OSS}$, and by $\rho^*_\text{OSS}(s'_0, s'_n)$ a goal reaching path from $s'_0 = C_\Gamma(s_0)$ to a goal state $s'_n \in \mathcal{S}_{s_\star}$, we call a path $\rho_\text{OSS}$ in $G_\text{OSS}$ surrogate path, and a sequence of operator labels in $\mathcal{T}'_\Pi$ surrogate plan, denoted by $\pi'$. OSS terminates search when it reaches a goal state $t \in \mathcal{S}_{s_\star}$ since $\Gamma$ stabilizes the goal. The final step is tracing back the optimal surrogate path $\rho^*_\text{OSS}(C_\Gamma(s_0), t)$ in the search tree $T_\text{OSS}$ to obtain a surrogate plan $\pi'$, and tracing forward surrogate path to decode an optimal plan $\pi$ from $\pi'$. We illustrate the final decoding step for the gripper task in Figure 1.

**Example 3 (TRACE-FORWARD in Gripper Task)** *A plan, corresponding to a black trace from the initial state, is extracted from a surrogate plan, which corresponds to a blue trace in Figure 1. These red nodes represent canonical states, while black nodes are the actual states obtained by applying the actions, when different from the canonical. Structural symmetries $\sigma_1$, $\sigma_2$, and $\sigma_3$ are obtained from the canonical mappings and are used to map actions on the trace into an applicable sequence of actions that is a plan.*

## Orbit Space $K^*$ Search

In this section, we present the orbit space $K^*$ search algorithm (O$K^*$) for top-$k$ planning, which alternates OSS and $EA$. We start by presenting a symmetry preserving single-goal state reformulation since the earlier reformulation breaks structural symmetries. Then, we show the equivalence between plans of given planing task $\Pi$ and surrogate plans reconstructed in $EA$, which is the basis for the soundness and completeness of O$K^*$ search algorithm for top-$k$ planning. We conclude this section by highlighting the difference between $K^*$ and O$K^*$ search algorithms.

### Symmetry and Single Goal State Reformulation

Some structural symmetries for the input planning task $\Pi$ can disappear in a single goal state reformulated task $\Pi^g$. Consider a reformulated gripper task following Katz et al. (2018b), where its single goal state $s^g_\star$ inherits facts in an arbitrary full state in $\Pi$. It is trivial to check a structural symmetry permuting ball $b_1$ and $b_2$ shown in Example 2 fails to stabilize $s^g_\star$ if $\{\langle b_1, \text{A}\rangle, \langle b_2, \text{B}\rangle\} \subseteq s^g_\star$. In order to preserve the structural symmetries of $\Pi$, we propose a new single goal state reformulation.

**Proposition 1** *A single goal state planning task $\Pi^g = \langle \mathcal{V}^g, \mathcal{O}^g, s^g_0, s^g_\star\rangle$ can be obtained from a planning task $\Pi = \langle \mathcal{V}, \mathcal{O}, s_0, s_\star\rangle$ as follows.*

- $\mathcal{V}^g = \{\overline{v} \mid v \in \mathcal{V}\} \cup \{v_g\}$, *where* $dom(\overline{v}) = dom(v) \cup \{\text{U}\}$ *and* $dom(v_g) = \{\text{I}, \text{U}\}$,
- $s^g_0 = s_0 \cup \{\langle v_g, \text{I}\rangle\}$,
- $s^g_\star = \{\langle v_g, \text{U}\rangle\} \cup \{\langle v, \text{U}\rangle \mid v \in \mathcal{V}\}$,

- $\mathcal{O}^g = \{\langle pre(o) \cup \{\langle v_g, \text{I}\rangle\}, \text{eff}(o)\rangle | o \in O\} \cup \{o_g\}$, *where* $pre(o_g) = s_\star \cup \{\langle v_g, \text{I}\rangle\}$, $\text{eff}(o_g) = s^g_\star$, *and* $cost(o_g) = 0$. *The cost of existing actions remains the same.*

Note that the additional goal-achieving zero-cost action $o_g$ can be applied only once when the original goal was achieved. No action is applicable in the new goal state, and therefore there is one-to-one correspondence between the plans of $\Pi$ and those of $\Pi_g$.

**Theorem 1** *Given a planning task $\Pi = \langle \mathcal{V}, \mathcal{O}, s_0, s_\star\rangle$ and the reformulated task $\Pi^g = \langle \mathcal{V}^g, \mathcal{O}^g, s^g_0, s^g_\star\rangle$ shown in Proposition 1, there exists a bijective mapping between structural symmetries of $\Pi$ and of $\Pi^g$.*

**Proof:** A structural symmetry $\sigma^g$ of $\Pi^g$ stabilizes $s^g_\star$ by definition, and it also stabilizes $o_g$ since $\text{eff}(o_g) = s^g_\star$. Thus, $\langle v_g, \text{I}\rangle$ in $pre(o_g)$ is mapped to itself, and so is the fact $\langle v_g, \text{U}\rangle$. $\sigma^g$ restricted to $\mathcal{V}$, $F$, and $\mathcal{O}$ is a structural symmetry of $\Pi$. In the other direction, let $\sigma$ be a structural symmetry of $\Pi$. We extend $\sigma$ to $\sigma^g$ of $\mathcal{V}^g$, $F^g$, and $\mathcal{O}^g$ as follows. The variable $v_g$ and its facts $\langle v_g, \text{I}\rangle$ and $\langle v_g, \text{U}\rangle$, as well as the action $o_g$ are mapped to themselves. Each extra fact $\langle v, \text{U}\rangle$ for the variable $v$ other than $v_g$ is mapped to $\langle \sigma(v), \text{U}\rangle$. It is trivial to see that $\sigma^g$ satisfies all properties of the structural symmetries in Definition 2. $\square$

### Plans in State Space and Orbit Space

OSS has shown its practical merit for cost-optimal planning (Alkhazraji et al. 2014; Sievers and Katz 2018; Katz et al. 2018a). The computation gain compared to $A^*$ stems from the fact that OSS explores a significantly smaller search space when symmetries are detected. In cost-optimal planning, or top-$k$ planning with $k = 1$, it is sufficient for only one optimal plan to be preserved while pruning symmetric states. For top-$k$ planning in general, however, all plans need to be preserved while pruning symmetric states.

**Proposition 2** *Given a planning task $\Pi = \langle \mathcal{V}, \mathcal{O}, s_0, s_\star\rangle$, let $P$ denote the set of all plans in the state transition graph $\mathcal{T}_\Pi$, and $P'$ denote the set of all surrogate plans in the canonical state transition graph $\mathcal{T}'_\Pi$ relative to a structural symmetry group $\Gamma$ of the planning task $\Pi$. Then, there exists a bijective mapping between plans in $P$ and surrogate plans in $P'$.*

**Proof:** We define a bijective mapping $\Phi$ between plan traces (state action sequences) in $\mathcal{T}_\Pi$ and in $\mathcal{T}'_\Pi$, from which the desired mapping can be straightforwardly extracted. Given a plan $\pi = \langle o_1, \ldots, o_n\rangle$ in $P$ that traverses a sequence of states $\langle s_0, s_1, \ldots, s_n\rangle$ over the state transition graph $\mathcal{T}_\Pi$, let us consider a sequence $\phi = \langle s_0, o_1, s_1, \ldots, o_n, s_n\rangle$, and a mapping $\Phi$ that maps $\phi$ to $\phi' = \langle s'_0, o'_1, s'_1, \ldots, o'_n, s'_n\rangle$ as follows.

- $s'_0 = C_\Gamma(s_0)$ and $\sigma_0$ is such that $\sigma_0(s_0) = C_\Gamma(s_0)$,
- $o'_1 = \sigma_0(o_1)$,
- $s'_i = C_\Gamma(s'_{i-1}[\![o'_i]\!])$ and $\sigma_i$ is such that $\sigma_i(s'_{i-1}[\![o'_i]\!]) = C_\Gamma(s'_{i-1}[\![o'_i]\!])$, and
- $o'_{i+1} = \sigma_{0:i}(o_{i+1})$, where $\sigma_{0:i} := \sigma_i \circ \sigma_{i-1} \circ \cdots \circ \sigma_0$.

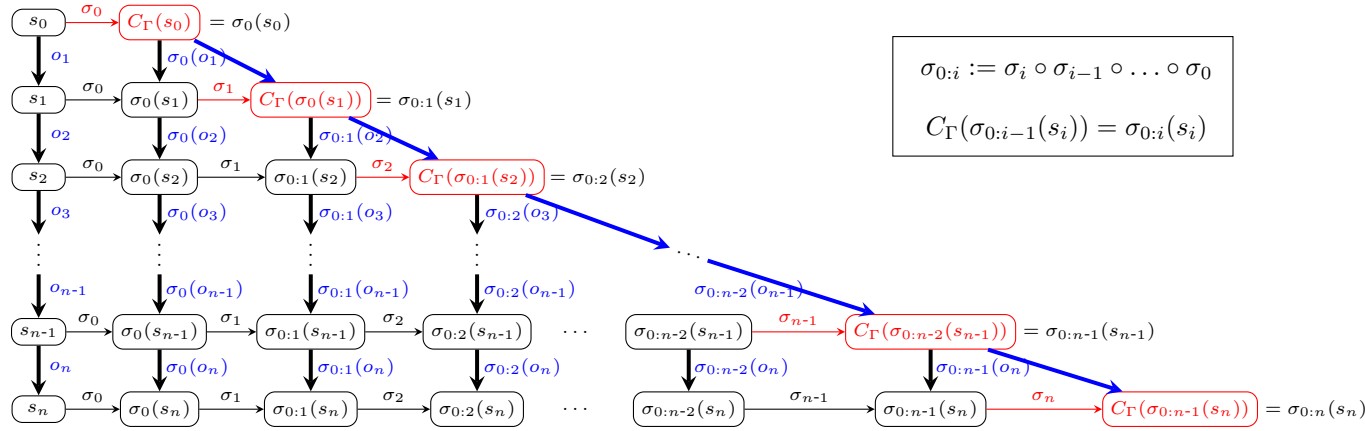

Figure 2: The visualization of the mapping between the plan $\pi = o_1 \ldots o_n$ and a surrogate plan.

Given that, we can also derive that $s'_i = \sigma_{0:i}(s_i)$. Figure 2 depicts the mapping described above. The plan $\pi$ and its corresponding sequence $\phi$ on the left is mapped to the sequence $\phi'$ of red states and blue action edges, which corresponds to the surrogate plan $\langle o'_1, \ldots, o'_n \rangle$. To see that $\phi'$ corresponds to a surrogate plan, observe that (i) each state $\sigma_{0:i}(s_i)$ is a canonical state, (ii) the action $\sigma_{0:i}(o_{i+1})$ is applicable in $\sigma_{0:i}(s_i)$ since $o_{i+1}$ is applicable in $s_i$, and (iii) applying $\sigma_{0:i}(o_{i+1})$ in $\sigma_{0:i}(s_i)$ results in a state $\sigma_{0:i}(s_i[\![o_{i+1}]\!]) = \sigma_{0:i}(s_{i+1})$, whose canonical state is $\sigma_{0:i+1}(s_{i+1})$. Finally, since $s_n$ is a goal state, so is $s'_n = \sigma_{0:n}(s_n)$. We show now that $\Phi$ is injective. For sequences $\phi^1 = \langle s_0^1, o_1^1, s_1^1, \ldots, o_n^1, s_n^1 \rangle$ and $\phi^2 = \langle s_0^2, o_1^2, s_1^2, \ldots, o_n^2, s_n^2 \rangle$ with $s_0^1 = s_0^2 = s_0$, if $\Phi(\phi_1) = \Phi(\phi_2)$, then we show by induction that the two sequences must be equivalent. Note that $\Phi(\phi^1)$ can be written as $\langle \sigma_0^1(s_0), \sigma_0^1(o_1^1), \sigma_{0:1}^1(s_1^1), \ldots, \sigma_{0:n-1}^1(o_n^1), \sigma_{0:n}^1(s_n^1) \rangle$, and $\Phi(\phi^2)$ in a similar way. Base: $s_0^1 = s_0^2 = s_0$ and $\sigma_0^1(s_0) = \sigma_0^2(s_0) = C_\Gamma(s_0)$. Therefore $\sigma_0^1 = \sigma_0^2$. Step: if $s_i^1 = s_i^2$ and $\sigma_i^1 = \sigma_i^2$ for $i \le m$, we show that $o_{m+1}^1 = o_{m+1}^2$ and $s_{m+1}^1 = s_{m+1}^2$. From the equivalence of $\Phi(\phi^1)$ and $\Phi(\phi^2)$, we have $\sigma_{0:m}^1(o_{m+1}^1) = \sigma_{0:m}^2(o_{m+1}^2)$. Since $\sigma_{0:m}^1 = \sigma_{0:m}^2$, we must have $o_{m+1}^1 = o_{m+1}^2$. Since $\phi^1$ and $\phi^2$ correspond to plans, we have $s_{m+1}^1 = s_m^1[\![o_{m+1}^1]\!]$ and $s_{m+1}^2 = s_m^2[\![o_{m+1}^2]\!]$, giving us $s_{m+1}^1 = s_{m+1}^2$.

We conclude by showing that $\Phi$ is surjective. For a surrogate plan $\pi' = \langle o'_1, \ldots, o'_m \rangle$ and its corresponding sequence $\phi' = \langle s'_0, o'_1, s'_1, \ldots, o'_n, s'_n \rangle$, we have $s'_0 = C_\Gamma(s_0)$ and $s'_i = C_\Gamma(s'_{i-1}[\![o'_i]\!])$. Let $\sigma_0$ be such that $\sigma_0(s_0) = s'_0 = C_\Gamma(s_0)$ and for $1 \le i \le n$, let $\sigma_i$ be such that $\sigma_i(s'_{i-1}[\![o'_i]\!]) = s'_i$. Let $o_i = \sigma_{0:i-1}^{-1}(o'_i)$ be the actions that are obtained by applying the inverse of the structural symmetry $\sigma_{0:i-1}$ to $o'_i$. We show that $\pi = \langle o_1, \ldots, o_n \rangle$ is a plan and for its corresponding sequence $\phi$ we have $\Phi(\phi) = \phi'$. Let $s_i = \sigma_{0:i}^{-1}(s'_i)$. First, observe that $s_0 = \sigma_0^{-1}(s'_0)$. We need to show that (i) $o_i$ is applicable in $s_{i-1}$ and (ii) applying $o_i$ in $s_{i-1}$ results in $s_i$. For (i) we know that $o'_i$ is applicable in $s'_{i-1}$ and for any $\sigma \in \Gamma$, $\sigma(o'_i)$ is applicable in $\sigma(s'_{i-1})$, and in particular for $\sigma_{0:i-1}^{-1}$.

**Algorithm 1** $OK^*$ Search

---
**Require:** Reformulated planning task $\Pi^g$, $k$
**Ensure:** Top-$k$ solution
1: Initialize OSS search
2: $P \leftarrow \emptyset$
3: **while** True **do**
4:     **while** $\neg(\text{OPEN}_{\text{OSS}} = \emptyset \vee \text{SWITCH-TO-}EA(\,))$ **do**
5:         Expand an OSS node
6:     PREPARE$EA()$
7:     **while** $\neg(\text{OPEN}_{EA} = \emptyset \vee \text{SWITCH-TO-OSS}(\,))$ **do**
8:         Expand an $EA$ node $n$
9:         Reconstruct surrogate plan $\pi'$ from $n$
10:         $P \leftarrow P \cup \{\text{TRACE-FORWARD}(\pi')\}$
11:         **if** $|P| = k$ **then return** $P$
12:     **if** $\text{OPEN}_{\text{OSS}} = \emptyset \wedge \text{OPEN}_{EA} = \emptyset$ **then return** $P$

---

For (ii), $s'_i = \sigma_i(s'_{i-1}[\![o'_i]\!])$ and therefore $\sigma_i^{-1}(s'_i) = s'_{i-1}[\![o'_i]\!]$. Now, $s_{i-1}[\![o_i]\!] = \sigma_{0:i-1}^{-1}(s'_{i-1})[\![\sigma_{0:i-1}^{-1}(o'_i)]\!] = \sigma_{0:i-1}^{-1}(s'_{i-1}[\![o'_i]\!]) = \sigma_{0:i-1}^{-1}(\sigma_i^{-1}(s'_i)) = \sigma_{0:i}^{-1}(s'_i) = s_i$. $\qquad\square$

### Implicit Path Representation in Orbit Space

Given a single goal state reformulated planning task $\Pi^g = \langle \mathcal{V}^g, \mathcal{O}^g, s_0^g, s_\star^g \rangle$, $EA$ in $K^*$ builds path graph $P(G_{A^*})$ from an explicit search graph $G_{A^*}$ of $A^*$ by exploring $\mathcal{T}_{\Pi^g}(\mathcal{S}, \mathcal{E})$. When $EA$ traverses a path graph $P(G_{\text{OSS}})$ derived from an explicit search graph $G_{\text{OSS}}$ of OSS, the implicit path representation of $EA$ is defined relative to the canonical state transition graph $\mathcal{T}'_{\Pi^g}(\mathcal{S}', \mathcal{E}')$. Nevertheless, for any goal reaching surrogate path $\rho_{\text{OSS}}(C_\Gamma(s_0^g), s_\star^g)$, the implicit path representation $\text{SIDETRACKS}(\rho_{\text{OSS}}(C_\Gamma(s_0^g), s_\star^g))$ still holds because $G_{\text{OSS}}$ is merely another digraph. Therefore, any path graph node generated by $EA$ in orbit space can be safely reconstructed to a surrogate path, which in turn, decoded back to a plan in the state space as shown in Proposition 2.

### Orbit Space $K^*$ Search Algorithm

Algorithm 1 shows $OK^*$ search algorithm for top-$k$ planning, which takes a single goal state reformulated task $\Pi^g$

| | | SymK | FI | LMcut | | M&S | | | CEGAR | | iPDB | | BLIND | |
|---|---|---|---|---|---|---|---|---|---|---|---|---|---|---|
| | | | | $K^*$ | $OK^*$ | $^{(+)}K^*$ | $K^*$ | $OK^*$ | $K^*$ | $OK^*$ | $K^*$ | $OK^*$ | $K^*$ | $OK^*$ |
| | SymK | 0 | **36** | **27** | 20 | **33** | **34** | **28** | **36** | **31** | 30 | 25 | **46** | **37** |
| | FI | 20 | 0 | 19 | 6 | 29 | 30 | 23 | 27 | 20 | 21 | 14 | **31** | 26 |
| LMcut | $K^*$ | 26 | **37** | 0 | 1 | **32** | **34** | **28** | 30 | **30** | 17 | 15 | **46** | **40** |
| | $OK^*$ | **32** | **45** | **30** | 0 | **45** | **47** | **35** | **45** | **36** | 30 | 19 | **57** | **50** |
| M&S | $^{(+)}K^*$ | 17 | 29 | 15 | 8 | 0 | **6** | 2 | **25** | 16 | 11 | 8 | **41** | 29 |
| | $K^*$ | 17 | 30 | 12 | 6 | 2 | 0 | 0 | 22 | 12 | 7 | 5 | **41** | 27 |
| | $OK^*$ | 20 | **34** | 23 | 12 | 22 | 24 | 0 | 31 | **23** | 20 | 9 | 48 | **41** |
| CEGAR | $K^*$ | 19 | **31** | 11 | 9 | 20 | 22 | 20 | 0 | 8 | 10 | 10 | **39** | 28 |
| | $OK^*$ | 23 | **34** | 22 | 10 | **30** | **32** | 22 | 25 | 0 | 23 | 9 | 48 | 33 |
| iPDB | $K^*$ | 30 | **38** | **22** | 19 | **35** | **36** | **32** | **32** | **28** | 0 | 1 | 48 | 38 |
| | $OK^*$ | **34** | **41** | **33** | 25 | **44** | **47** | **35** | **41** | **36** | 22 | 0 | 54 | 46 |
| BLIND | $K^*$ | 11 | 28 | 3 | 1 | 7 | 7 | 7 | 2 | 1 | 3 | 2 | 0 | 0 |
| | $OK^*$ | 15 | **31** | 14 | 3 | 17 | 17 | 7 | 19 | 2 | 12 | 2 | **29** | 0 |
| Overall Coverage | | 892 | 652 | 933 | **1022** | 786 | 769 | 834 | 803 | 868 | 862 | 932 | 587 | 695 |

Table 1: Pair-wise domain level comparison of top-k planners for $k = 1000$. Each entry in the table represents the number of domains where the row configuration achieves better coverage than the column one. The last row depicts the overall coverage.

and $k$ as input. Initialization steps for OSS detects structural symmetries of $\Pi^g$ and creates necessary data structure to explore the canonical state transition graph $\mathcal{T}'_{\Pi^g}$ (line 1). After initializing $P$ to store found plans (line 2), $OK^*$ alternates OSS (line 4–5) and $EA$ (line 7–11). OSS explores $\mathcal{T}'_{\Pi^g}$ until it exhausts the search space or SWITCH-TO-$EA$() triggers to stop expanding nodes in OSS. The latter happens when either the lowest f value in the OSS queue is no smaller than the one in the EA queue, or a pre-defined threshold on the number of expanded nodes since previous switch is reached. Before initiating $EA$, PREPARE$EA$() builds necessary data structures such as OPEN$_{EA}$ and heap graphs (Eppstein 1998). $EA$ traverses the path graph of the sub-graph of $\mathcal{T}'_{\Pi^g}$ developed by OSS. For a path graph node, $OK^*$ first reconstructs surrogate plan $\pi'$, and then decodes it to a plan $\pi$ by TRACE-FORWARD (Domshlak, Katz, and Shleyfman 2015). If the lowest f value in the OSS queue is smaller than the one in the EA queue, SWITCH-TO-OSS() triggers, forcing a switch back to OSS. All the changes from the $K^*$ algorithm are marked in blue. $OK^*$ terminates either when it finds top-$k$ plans for a solvable top-$k$ problem (line 11) or it exhausts both open lists before finding $k$ plans for an unsolvable top-$k$ problem (line 12).

**Theorem 2** *Algorithm 1 ($OK^*$) is sound and complete for top-k planning.*

**Proof:** Let $\Pi$ be a planning task and let $P \subseteq \mathcal{P}_\Pi$ be the set of plans found by Algorithm 1 for k. We show that $P$ is a solution to top-k planning problem. First, if $|P| < k$, then the algorithm terminated in line 12. Thus, it has exhausted the orbit space and reconstructed plans that correspond to all surrogate plans in the orbit space. From Proposition 2 we have an isomorphism between the (finite) set of surrogate plans in the orbit space and $\mathcal{P}_\Pi$, the set of plans of $\Pi$. Therefore, we must have $P = \mathcal{P}_\Pi$.

Now, suppose the algorithm terminated in line 11 and let $\rho'$ be the surrogate path extracted last before the termination. Then for all $\pi \in P$, we have $cost(\pi) \leq cost(\rho')$,

since $EA$ nodes are expanded in the order of their costs. Assume to the contrary that there exists a plan $\pi \notin P$, such that $cost(\pi) < cost(\pi')$ for some $\pi' \in P$. Then, according to Proposition 2, let $\rho$ be the surrogate plan that corresponds to $\pi$. Since $\pi \notin P$, $EA$ did not expand the node that corresponds to $\rho$ yet. However, note that we have $cost(\rho) < cost(\rho')$, contradicting the expansion order of $EA$. $\square$

## Experimental Evaluation

To empirically evaluate the effectiveness of using structural symmetries with $K^*$, we have implemented our suggested algorithm $OK^*$ on top of an existing $K^*$ implementation (Lee, Katz, and Sohrabi 2023) within the Fast Downward planning system (Helmert 2006). All experiments were performed on Intel(R) Xeon(R) Gold 6248 CPU @ 2.50GHz machines, with the timeout of 30 minutes and memory limit of 8GB per run. The benchmark set consists of all benchmarks from optimal tracks of International Planning Competitions 1998-2018, a total of 1827 tasks in 65 domains. We have experimented with four admissible heuristics, LMcut (Helmert and Domshlak 2009), merge-and-shrink abstraction (denoted by M&S) (Helmert, Haslum, and Hoffmann 2007), counterexample-guided Cartesian abstraction refinement (denoted by CEGAR) (Seipp and Helmert 2018), and pattern database heuristic iPDB (Haslum et al. 2007), as well as with the blind heuristic. We measure the total time for finding the top-k solution for a given $k$, as well as the coverage: 1 per task if the top-k solution was found, 0 otherwise. In addition to comparing $OK^*$ to $K^*$ with the aforementioned heuristics, we compare to the planners that implement the other two approaches to top-k planning, bi-directional symbolic search (SymK) (Speck, Mattmüller, and Nebel 2020) and Forbid Iterative (FI) (Katz et al. 2018b). Following previous work, all planners are using the same translator and no additional preprocessing, as the preprocessing can remove some actions that could otherwise be part of valid

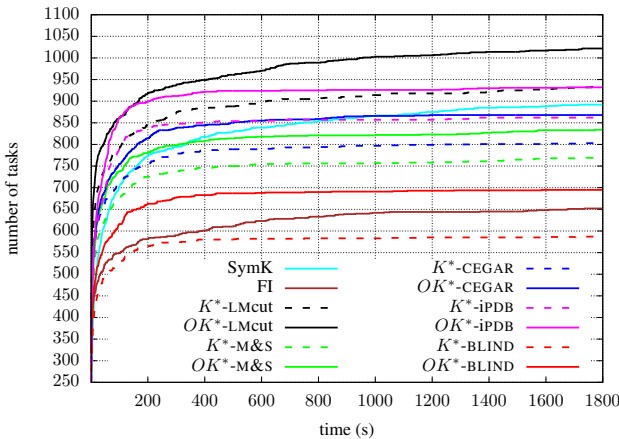

Figure 3: Anytime performance of selected configurations.

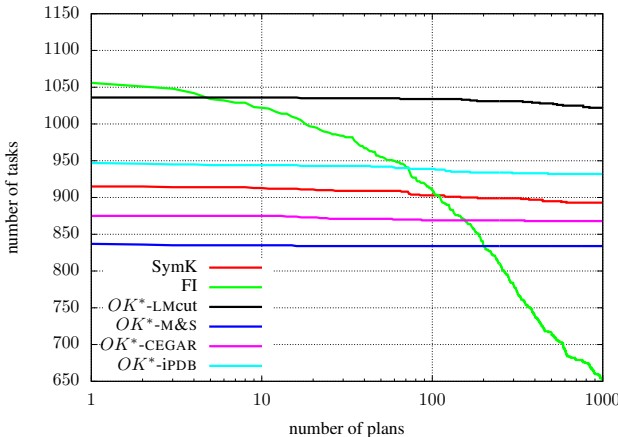

Figure 4: Any-k performance of selected configurations.

plans. The merge-and-shrink abstraction performs an optimization, pruning abstract states that are unreachable from the (abstract) initial state. This optimization must be turned off when using orbit space search. In our experiments, we compare to both with and without the optimization when using $K^*$. To differentiate the two, we mark the configuration with the optimization by $^{(+)}$.

Table 1 shows the pairwise comparison of the tested approaches for $k = 1000$. Each entry denotes the number of domains where the row planner achieves a better summed coverage than the column planner. Additionally, the last row denotes the overall coverage for each planner. The winners are marked in bold. For the pairwise comparison, the value in (x,y) is bolded if it is larger than the value in (y,x), that is planner x excelled over planner y in more domains than planner y excelled over planner x. First, note that there are a few instances solved by $K^*$ but not by $OK^*$, specifically 25 for the CEGAR heuristic and one for each of the other heuristic. While a few of these instances are due to minor time fluctuations, the rest appear to be due to (abstraction based) heuristics not being as informative on canonical states as on states reachable from the initial state. Despite that shortcoming, we observe that $OK^*$ consistently significantly outperforms $K^*$. The largest overall coverage increase occurs for the blind heuristic, 108 tasks. Next is LMcut with 89 tasks, iPDB with 70 tasks, and CEGAR and M&S with 65 tasks each. The best performing overall configuration is $OK^*$ with LMcut heuristic, passing SymK by 130 tasks overall. Looking at domain level performance, LMcut wins over SymK in 32 domains and loses in 20, showing superior performance. Comparing to other configurations on a domain level coverage, LMcut almost always wins in more domains than it loses in. The only exception is $OK^*$ with iPDB, which is the best domain level performer, always winning in more domains than it is losing in and winning over LMcut in 25 domains, loosing in 19.

As many planning applications might require a shorter planning time, we test the any-time performance of $OK^*$ compared to the other approaches. The results for top-k cov-

erage for $k = 1000$ as a function of time are depicted in Figure 3. The lines for the same heuristic are depicted with the same color, solid line for $OK^*$ and dashed line for $K^*$. Our first observation is that for all heuristics and all timeouts, the configurations that run $OK^*$ significantly outperform the $K^*$ configurations. The overall best performer for almost all tested timeouts is $OK^*$ with LMcut heuristic. Only between 100 and 150 seconds it loses the first place to $OK^*$ with iPDB. For these timeouts, the difference in overall coverage between the two approaches peaks at 6 tasks. Somewhat surprisingly, for small timeouts of up to 45 seconds, the second best is $K^*$ with LMcut. From 50 seconds to 1700 seconds, $OK^*$ with iPDB is the second best. From 1700 till 1760 it is tied with $K^*$ with LMcut, which takes the lead for the last 40 seconds. It is worth mentioning that $OK^*$ with iPDB achieves almost its maximal coverage (922 out of 932) and levels out in under 400 seconds. The coverage of $OK^*$ with LMcut, on the other hand, keeps rising up with more allocated time.

Finally, to test the performance of our approach for smaller $k$ values, Figure 4 depicts the top-k coverage as a function of k. Note here that for very small values of $k$, FI still outperforms all other approaches. For $k = 1$, as in cost-optimal planning, FI achieves the overall coverage of 1056 vs. 1036 for $OK^*$ with LMcut. Since FI uses internally OSS with LMcut and partial order reduction, and $OK^*$ for $k = 1$ is essentially equivalent to OSS, the difference stems from our methods not being able yet to benefit from partial order reduction. As FI needs to iteratively run a cost-optimal planner, it quickly loses its benefit and $OK^*$ with LMcut takes the lead already for $k = 5$ and keeps the first place for all $k \geq 5$. FI keeps the second place until $k = 72$, where $OK^*$ with iPDB catches up, keeping the second place from that point onward. SymK takes the third place starting from $k = 107$. It is worth noting that $OK^*$ configurations (as well as SymK and $K^*$) do not lose much coverage when going to larger $k$ values, allowing to generate many plans quickly.

## Conclusions and Future Work

In this work, we exploit structural symmetries to improve the performance of $K^*$ search. For that, we propose a new single-goal planning task transformation that preserves the symmetries of the input task. We formally prove a property of the orbit space that allows us to apply $K^*$ to it. We show that the proposed algorithm is sound and complete for top-k planning and perform an experimental evaluation that establishes our approach as the new state of the art for top-k planning.

In the future work we intend to further improve $OK^*$ for top-k planning by integrating another pruning technique, partial order reduction (Wehrle and Helmert 2012). In contrast to symmetry based pruning, applying partial order reduction does prune some of the goal paths in the search space (Katz and Lee 2023), and therefore great care must be taken to ensure that all plans can be found. Another promising direction is integrating symmetries explicitly into Eppstein's algorithm, to reduce both the computational effort and the memory consumption required for storing all paths by storing symmetric paths within the same node.

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
