# OpenReview forum: "K* Search Over Orbit Space for Top-k Planning"
_icaps-conference.org/ICAPS/2023/Workshop/HSDIP — ICAPS HSDIP 2023_

### Official Review · Reviewer_fpua · 2023-04-25
**Nice work!**

**Rating:** 7
**Confidence:** 4

**Review:**

The paper introduces symmetry breaking via orbit-space search (OSS) to the K* algorithm for top-K planning. The K* algorithm is designed to find plans in order of increasing cost by doing a Dijkstra-like search on the search space explored by A* for a given planning task. Orbit search has been introduced in classical planning to prune the search space in an optimality-preserving way. It does so by replacing every generated state by its canonical representative, which is obtained by applying the structural symmetries of the task.
As K* requires a search space with a single goal state, the authors introduce a transformation to the original task that does that, while preserving all structural symmetries. The authors then show that K* can still extract all solutions from the orbit search space.
In some experimental evaluation, the new approach compares favorably against other state-of-the-art top-K planners.

The paper is a clear fit for the workshop and describes interesting results in top-K planning.


Minor:
- intro: "first time using inconsistent heuristics for top-k planning". That's not quite true, is it? Inconsistent heuristics were previously usable in forbid-iterative, just not in K*.
- type in the header of example 2
- Definition 4, 2.: there is one \sigma_j that should not be there and the last \sigma_j should probably be \sigma_i
- Orbit Space K* Search: "symmetreis"
- the cite of Katz et al. [2018b] uses brackets, where all other cites use parentheses for the year.
- Proposition 1 looks more like a definition
- Algorithm 1: "exapnded"
- the closing parenthesis of the Switch-to-EA and PrepareEA functions have wrong style in the plain text.

---

### Official Review · Reviewer_UHpk · 2023-04-26
**Successful combination of known techniques to speed up K* with opportunities to clarify presentation.**

**Rating:** 7
**Confidence:** 2

**Review:**

### Summary
Top-$k$ planning considers the problem of finding a set of $k$ plans
such that no cheaper plans exist that are not in this set. This paper
claims to improve the state-of-the-art in this field. Their approach to
top-$k$ planning is an improved version of $K^*$ search over orbit
spaces. $K^*$ repeatedly performs $A^*$ search and therefore depends on
performance of $A^*$. Orbit spaces exploit symmetries in a given
planning problem to prune certain aspects during search. While this
technique was successfully applied to cost-optimal planning where the
goal is to find exactly one cheapest plan, it has not been used for
top-$k$ planning yet. The main contribution of this paper is filling
this gap and an empirical study shows that it pays off to do so.

The paper combines known techniques in a new context and therefore
novelty is limited. Nevertheless, the paper fits the scope of HSDIP
well, the technique works well in practice, and the contribution is
interesting. These reasons are enough to justify acceptance for the
workshop in my opinion. I also did not find any fundamental flaws, but I
did have a hard time following the theory in some parts. I believe the
paper could be made more accessible by clearer notation, see some
suggestions below. Furthermore, I think the paper is not yet
self-contained (e.g., I still have no intuition about how Eppstein's
algorithm works).

The experimental evaluation is in my opinion a weak part of the paper.
Although there is evidence that the newly proposed technique for top-$k$
planning works well in practice, I find little insight in why this is
the case. See some questions below (Question 2).

### Suggestions to Clarify Presentation
- Introduction: Order previous approaches to top-$k$ planning
  chronologically.
- Background; Top-$k$ Classical Planning: $s_0$ is overloaded in the
  definition of applicable action sequences.
- Background; Implicit Path Representation of Eppstein's Algorithm:
  Definition of edges induced by $\mathcal{O}$: $\bigcup_{s \in
  \mathcal{S}} \{\langle s, s[o]; o\rangle \mid o \in \mathcal{O},
  \textit{pre}(o) \subseteq s\}$ (curly brackets not displayed for some
  reason)
- Definition 2: Start by saying that you abuse notation of $\sigma$ for
  subsets of $\mathcal{V} \cup F \cup \mathcal{O}$ before listing all
  the properties.
- Colors of Figures are not distinguishable in BW print.
- Table 1: Group results in $K^*$ and $OK^*$. Also, replace 0 on the
  diagonal by something like -- so that the diagonal more clearly
  separates upper and lower triangle. In the current version, I cannot
  see any systematic behaviour, but this should be the goal of
  presenting results in this way in my opinion.

### Questions
(1) There is a reference to an unpublished submission to SoCS called
"On $K^*$ Search for Top-$k$ Planning". What is the difference of this
paper to the SoCS submission?

(2) Questions about experiments:

  (a) Why must the merge-and-shrink pruning optimization be turned off
  for orbit space search?

  (b) Did you consider why $K^*$ works better than $OK^*$ in some
  domains? Which domains? Any interpretations?

  (c) Do you have any ideas why LM-Cut and iPDB beat SymK while CEGAR
  and M&S don't?

  (d) What do you read out of the coverage difference for individual
  heuristics between $K^*$ and $OK^*$? Simply stating them is not very
  interesting in my opinion.

  (e) The paragraph about anytime performance of $OK^*$ confuses me a
  lot. I cannot figure out much meaning from the many numbers dropped in
  this paragraph.

### Minor Comments
#### General
- Consistently use top-$k$ rather than top-k.

#### Background
- Definition of successor state: "for all $v \in
  \textit{vars}(\textit{eff})$" --> "for all $v \in
  \textit{vars}(\textit{eff}(o))$"
- Example 1, bullet point 2: "$\langle b_1,r \rangle$, $\langle b_2,l
  \rangle$" --> "$\langle b_1,R \rangle$, $\langle b_2,R \rangle$"
- Paragraph after Definition 4: $\mathcal{S}_{s_*}$ is undefined.

#### Orbit Space $K^*$ Search
- First sentence: "K*" -- "$K^*$".
- Second paragraph: Reference to Katz et al 2018b looks different.
- Proposition 2: "..., let's denote by $P$ a set ..." --> "..., let us
  denote by $P$ the set ..."
- Implicit Path Representation in Orbit Space: "When EA traverse ..."
  --> "When EA traverses ..."
- Orbit Space $K^*$ Search Algorithm: Formatting of closing brackets of
  textsc look off.

---

> ### Author Response · Authors · 2023-04-28
> **Unpublished submission to SoCS called "On K* Search for Top-k Planning"**
>
> The now accepted to SoCS paper titled "On K* Search for Top-k Planning" is about an implementation of K* search for classical planning. This work builds upon that implementation, adding symmetry based pruning to the variant of K* search presented in the SoCS paper.

---

> ### Author Response · Authors · 2023-04-28
> **Questions about experiments:**
>
> Thank you for insightful questions.
>
> (a) The M&S optimization in question removes abstract states that are either unreachable from the abstract initial state or the abstract goal is unreachable from them. For A*, this can be done, since no non-dead end states reached by A* would map to such abstract states. For orbit search, however, this is not the case, it can definitely evaluate states not reachable from the initial state.
>
> (b) There are a total of 28 cases when K* solved the task, while OK* failed. One for iPDB, LMcut and M&S, each, 25 for CEGAR.
> M&S: mystery:prob05.pddl, while the heuristic recognized the initial state as a dead end, the canonical of the initial state was not recognized as such, and the search was out of memory before proving unsolvable.
> LMcut: OK* was out of time, while K* managed to solve in 1500s.
> iPDB: OK* expands more nodes than K* and fails on memory.
> CEGAR: upon closer inspection, it seems like in these cases, CEGAR typically assigns lower heuristic values to canonical states.
> Previous work (Heuristics and Symmetries in Classical Planning, Shleyfman et al, AAAI 2015) has explored the invariance of heuristics under structural symmetries, but excluded abstractions from that investigation, leaving it to future work. It seems like such an investigation is now overdue.
>
> (c)  Non-optimized M&S is not as strong heuristic as it could have been. Similarly, CEGAR might be weaker than it could have been on canonical states. There are several ways to overcome these limitations. First, it might be possible to use DKS based K*, which does not have that limitation of OSS, but stores twice as many states. Second, an investigation into the interplay between symmetries and CEGAR heuristic might be in order.
>
> (d) We find it quite interesting to show the coverage difference between K* and OK*. We are open to suggestions on what kind of analysis readers would like to see.

---

> > ### Comment · Reviewer_UHpk · 2023-05-02
> > **Thanks for some more insights**
> >
> > I appreciate you taking the time to look into these questions and providing some more insights. I would also like to see some of these mentioned in the camera ready version of this paper.
> >
> > (b) I find the observation for M&S very interesting and it makes sense to me. Regarding LM-Cut: Do you have an idea why OK* takes longer for this instance? According to Figure 3, OK* generally solves tasks faster than K*, right? But if this is not consistently the case, I'm curious whether you can think of a reason why it is sometimes the other way around. For the Cartesian abstractions I would also be interested in more details, but I agree that this could be expanded on in its own paper.
> >
> > (d) If you find the numbers interesting, then I would assume you also have thought about why they might occur, e.g., why is the difference larger with LM-cut compared to abstraction heuristics. Spelling out these thoughts after stating the numbers would in my opinion increase the value of doing so, even if they are only hypotheses; at least I as a reader would appreciate that.

---

> > > ### Author Response · Authors · 2023-05-02
> > > **Re LMcut**
> > >
> > > In that particular case, the OSS time is comparable to the A* time, while the plan reconstruction step takes longer when surrogate paths are mapped to plans. This is the only case observed in our experiments when that made a difference between solving and not solving within the 30 minutes time bound.

---

> > > ### Author Response · Authors · 2023-05-02
> > > **why is the difference larger with LM-cut compared to abstraction heuristics**
> > >
> > > One obvious speculation is that if the heuristic takes longer to compute then saving on the same number of computations translates into a larger benefit. However, there are many things at play and things are rarely that simple.

---

### Decision · Program_Chairs · 2023-05-05

**Decision:**

Accept

**Comment:**

We are happy to inform you that this paper was accepted to be presented at HSDIP. Congratulations!

Please make sure to consider the comments made by the reviewers when preparing the final version. In particular, we ask you to expand the experimental section with observations you mention in the discussion below.